# Dilemma of physician-mothers faced with an increased home burden and clinical duties in the hospital during the COVID-19 pandemic

**Sachiyo Nishida**[1,2], **Kanna Nagaishi**[1,3°], **Masayo Motoya**[1,4°], **Ayako Kumagai**[1,5°], **Noriko Terada**[1,6°], **Ai Kasuga**[1,7°], **Narumi Kubota**[1,8°], **Kotoe Iesato**[1,7°], **Motonobu Kimizuka**[1,9°], **Satsuki Miyajima**[1,10°], **Masayuki Koyama**[11,12], **Hirofumi Ohnishi**[11], **Eichi Narimatsu**[8], **Naoya Masumori**[2], **Kazufumi Tsuchihashi**[12,13], **Taiji Tsukamoto**[14], **Yoshihisa Tsuji**[1,15]*

1 The Physicians' Career Support Committee, Sapporo Medical University, Sapporo, Japan, 2 Department of Urology, Sapporo Medical University, Sapporo, Japan, 3 Department of Anatomy, Sapporo Medical University, Sapporo, Japan, 4 Department of Gastroenterology and Hepatology, Sapporo Medical University, Sapporo, Japan, 5 Department of Dermatology, Sapporo Medical University, Sapporo, Japan, 6 Department of Gynecology, Sapporo Medical University, Sapporo, Japan, 7 Department of Pediatrics, Sapporo Medical University, Sapporo, Japan, 8 Department of Emergency Medicine, Sapporo Medical University, Sapporo, Japan, 9 Department of Anesthesiology, Sapporo Medical University, Sapporo, Japan, 10 Department of Respiratory Medicine, Sapporo Medical University, Sapporo, Japan, 11 Department of Public Health, Sapporo Medical University, Sapporo, Japan, 12 Department of Cardiovascular, Renal and Metabolic Medicine, Sapporo Medical University, Sapporo, Japan, 13 Division of Health Care Administration and Management, Sapporo Medical University, School of Medicine, Sapporo, Japan, 14 Department of University Administration, Sapporo Medical University, Sapporo, Japan, 15 Department of General Medicine, Sapporo Medical University, Sapporo, Japan

° These authors contributed equally to this work.
* ytsuji@sapmed.ac.jp

**Data Availability Statement:** All relevant data are within the manuscript and its S1 File, S1 Table files.

## Abstract

### Purpose

Since December 2019, coronavirus disease 2019 (COVID-19) has spread rapidly across the world. During the pandemic, physicians in our hospital have had to respond both to the issue of treating the patients and the increasing domestic burden associated with social disruption. The purpose of this study was to assess how much the burden on our doctors, especially female doctors, was increasing.

### Material and methods

The Physicians' Career Support Committee in Sapporo Medical University conducted a questionnaire survey. The questionnaire inquired about a wide range of subjects with regard to working style and family life during the first and second waves of the COVID-19 pandemic, and was sent to all medical/dental physicians working in Sapporo Medical University.

### Results

A total of 266 (42.7%) physicians in our hospital responded to our questionnaire and the data for 264 data were analyzed. The total numbers of males, females, and others, including those who did not want to specify, were 178 (67.4%), 82 (31.0%), and 4 (1.5%),

**Funding:** The authors received no specific funding for this work.

**Competing interests:** The authors have declared that no competing interests exist.

respectively. Among them, 62 (23.5%) and 23 (8.7%) answered that their domestic burden was slightly or markedly increased. The increase in the domestic burden showed a significant difference between genders (p = 0.04). Even after correction for background differences using multivariate analysis, being female (p<0.001), having child dependents (p<0.001), and treating COVID-19 patients (p = 0.03) were significantly related to an increased domestic burden. Regarding family style, 58.1% of the physician-fathers were from two-income families (i.e., families with both parents in employment), and they answered that their partner mainly cared for the children. In contrast, 97.3% of physician-mothers were from two-income families, and 94.6% of the physician-mothers had to take care of children by themselves.

## Conclusion

Physician-mothers are caught in a dilemma between an increased home burden and clinical duties in the hospital, with a significantly higher ratio than physician-fathers during the pandemic. As we showed, female doctors could have not continued their careers and take responsible positions in the same way as male doctors. This is a social risk in the timing of a crisis, such as a pandemic.

## Introduction

Since December 2019, coronavirus disease 2019 (COVID-19) has spread rapidly across the world. In Japan, the first confirmed case of SARS-CoV-2 infection was reported on January 16, 2020. The impact of COVID-19 on Hokkaido prefecture was also huge. From January through March (first wave) 2020, and from April through May (second wave) 2020, the number of COVID-19 patients in Hokkaido markedly increased. At that time, the local government decided that all schools and after-school care facilities in the Hokkaido area would be temporarily closed for more than one month and, after that, it continued to change school times according to the grade. This area had the first and most extended lockdown of any city in Japan during the first and second waves of the pandemic.

The area of Hokkaido is 83,423.84 km$^2$. It is the second-largest island of Japan, comprising 22% of Japan's total land area. It has the third-largest population of Japan's five main islands, with 5,383,579 people as of 2015. According to a summary by the Hokkaido Bureau of Economy, Trade and Industry, Hokkaido's regional gross domestic product (GDP) was 18.3 trillion yen in 2013; approximately the same as that of the 50[th] ranked country in the world in 2019. In Hokkaido, three medical universities educate medical students. Of these three universities, two are managed by the Japanese government and one by Hokkaido's prefectural government. As a core hospital in the city of Sapporo, Sapporo Medical University Hospital is required to treat patients with severe cases of COVID-19. On the other hand, as an institution that the prefectural government supports officially, it also dispatches many physicians to participate in rural healthcare services. According to a survey in 2020, 35.1% of the clinical fellows in Hokkaido's public hospitals were dispatched from Sapporo Medical University. It is no exaggeration to say that the physicians who belong to our hospital support Hokkaido's medical care.

As we indicated above, Sapporo Medical University is expected to contribute to Hokkaido's public health, and the contribution to rural healthcare is in accord with the institution's founding ethos. In actuality, according to the data regarding COVID-19 in Hokkaido, Department

of General Affairs, Hokkaido Government, during the first and second waves of the pandemic in Hokkaido, we treated 9.2% of the patients with COVID-19 in Hokkaido. Moreover, of all patients in Hokkaido, 42.1% of the severe cases and 75% of patients with veno-venous extra-corporeal membrane oxygenation were treated in our hospital, responding to the request of its governor. As well as clinical physicians, basic researchers and graduate students with medical doctor license were involved in the diagnosis of the COVID-19 suspected cases in the rural hospitals, too. Although we are proud of this fact, the burden on our hospital staff was increased. Importantly, during the pandemic, physicians in our hospital had to respond both to the issue of treating the patients and spending more time on household activities associated with social disruption. The current COVID-19 pandemic has been described to intensify workplace inequities for female physician [1], in this situation, there is especial concern that the burden of housework will be over-concentrated on female physicians and create a dilemma when faced with both internal and external duties.

For this reason, the Physicians' Career Support Committee in Sapporo Medical University performed a questionnaire survey of physicians working in our university to assess the domestic burden of the genders during the pandemic. From the results of the questionnaire, we analyzed the burden of housework, including the care of children mainly by gender.

## Materials and method

### Design and settings

The Physicians' Career Support Committee in Sapporo Medical University conducted this survey and made the online questionnaire. The site information (the URL) of the questionnaire was announced to all 622 medical and dental physicians and researchers, including 459 males and 163 females, working in Sapporo Medical University by e-mail on September 8, 2020. When the survey was conducted, it was announced for all physicians by e-mail and on our Japanese website that the results would be published on the web and the paper. As a participant answered the questionnaire, it was taken as an informed consent agreement for participation in this study. All of the responders, including researchers and graduate students, had a medical or dental physician's license and had the opportunity to take care of patients routinely.

The data collection period for this study was 20 days. To guarantee anonymity, no personal data that could allow the identification of respondents was included. This study was approved by the institutional reviewing committee of Sapporo Medical University (approval number 3-1-14).

### Questionnaire

Nine main items of the questionnaire were used in this study (S1 File). In the questionnaire, we asked about a wide range of subjects concerning working style and family life during the first and second waves of the COVID-19 pandemic. The questions were both open- and closed-ended, including information on demographic variables. These items and questions were decided in our committee meeting, referring to a national survey conducted by the Japanese Cabinet Office on June 21, 2020.

### Statistical analysis

Univariate analyses exploring relations among individual factors that increased the burden at home were performed using Fisher's exact test for two or larger-dimensional contingency tables. P<0.05 was considered significant. Multivariate linear regression analyses for the parameters that increased the burden at home were developed to explore the relative

contributions of the various factors. All statistical analysis was performed with EZR (Saitama Medical Center, Jichi Medical University, Saitama, Japan, version 1.42), which is a graphical user interface for R (The R Foundation for Statistical Computing, Vienna, Austria, version 4.0.0.) [2].

## Results

### Background information of responders to the questionnaire

Of the staff in our hospital, 266 (42.7%) responded to our questionnaire. Two of them were excluded because of data loss, so data for 264 responders were analyzed in this study (Table 1). The total numbers of males, females, and others, including those who did not want to specify, were 178 (67.4%), 82 (31.0%), and 4 (1.5%), respectively. Among these three gender categories, age distribution showed significant differences (p = 0.01). For example, females were markedly younger.

Of the responders, more than 50% of responders were attending physicians, followed by those with other positions. Regarding the distribution of employment positions in our hospital, there was a significant gender difference (p = 0.02). Males comprised 60.6% of the attending physicians, whereas 48.7% of the clinical fellows were female physicians.

Of all the responders, 160 (60.6%) had dependent relatives. Of these, 154 had children, 6 had adult children, 7 were pregnant mother, and 15 took care of elderly persons. Regarding the distribution of the dependent relatives, there was a significant difference between genders (p<0.01). Compared to male physicians, significantly fewer female physicians in our hospital answered that they had children as dependents (65.7 vs. 45.1%, p<0.01).

**Table 1. Characteristics of all responders in the questionnaire survey.**

| | | | Total (n = 264) | Males (n = 178) | Females (n = 82) | Others (n = 4) | p-value |
|---|---|---|---|---|---|---|---|
| Age (years) (%) | 20~29 | | 36 (13.6) | 13 (7.3) | 20 (24.3) | 3 (75.0) | 0.01[†] |
| | 30~39 | | 104 (39.4) | 71 (39.9) | 32 (39.0) | 1 (15.0) | |
| | 40~49 | | 87 (37.9) | 63 (35.3) | 24 (29.2) | 0 (0) | |
| | 50~59 | | 28 (10.6) | 24 (13.4) | 4 (9.5) | 0(0) | |
| | 60~69 | | 9 (3.4) | 7 (3.9) | 2 (2.4) | 0 (0)) | |
| Employment position (%) | Resident | | 15 (5.7) | 9 (5.0) | 5 (6.0) | 1 (15.0) | 0.02[†] |
| | Clinical fellow | | 70 (26.5) | 27 (15.1) | 40 (48.7) | 3 (75.0) | |
| | Researcher* | | 18 (6.8) | 11 (6.1) | 7 (8.5) | 0 (0) | |
| | Graduate student* | | 26 (9.8) | 23 (12.9) | 3 (3.6) | 0 (0) | |
| | Attending Physician | | 135 (51.1) | 108 (60.6) | 27 (32.9) | 0 (0) | |
| Dependent relative(s) (%) | No | | 104 (65.0) | 55 (30.9) | 45 (54.9) | 4 (100.0) | <0.01[†] |
| | Yes (n = 160, multiple answers) | children | 154 (58.3) | 117(65.7) | 37 (45.1) | 0 (0) | |
| | | adult children | 6 (2.3) | 6(3.4) | 0 (0) | 0 (0) | |
| | | pregnant | 7 (2.7) | 4 (2.2) | 3 (3.7) | 0 (0) | |
| | | elderly | 15 (56.8) | 3 (1.7) | 12 (14.6) | 0 (0) | |
| Experience of treating patients with COVID-19 (%) | Yes | | 134 | 100 (56.2) | 33 (40.2) | 1 (15.0) | 0.03[†] |
| | No | | 130 | 78 (43.8) | 49 (59.8) | 3(75.0) | |

*with Medical Doctor license, finished residency and engaged in treatment of COVID-19 patients.

[†]P<0.05 was considered significant.

Numbers in parentheses indicate percentages.

**Table 2. The results of the questionnaire survey regarding changes of domestic burden due to the pandemic.**

| Domestic burden before and after COVID-19 pandemic | Answers | Total (n = 264*) | Males | Females | Others | p-value |
|---|---|---|---|---|---|---|
| | | | (n = 178) | (n = 82) | (n = 4) | |
| | not increased/equivocal | 178 (67.4) | 127 (71.3) | 49 (59.8) | 2 (50.0) | 0.04[†] |
| | slightly increased | 62 (23.5) | 39 (21.8) | 21 (25.6) | 2 (50.0) | |
| | markedly increased | 23 (8.7) | 11 (6.2) | 12 (14.1) | 0 (0) | |

[†] P<0.05 was considered significant.

* There was a 20s male doctor who did not answer this question only, so the total number of respondents was 263.

Numbers in parentheses indicate percentages.

There was a significant difference between genders in the experience of treating patients with COVID-19 in our hospital (56.2% for males vs. 40.2% for females, p = 0.03).

## Increase of domestic burden associated with the pandemic according to gender

During the first and second waves of the COVID-19 pandemic, 62/264 (23.5%) and 23/264 (8.7%) answered that the domestic burden was slightly or markedly increased, respectively (Table 2). The increase was found for both genders. However, regarding a marked increase of the domestic burden associated with the pandemic, the ratio of female physicians with such an increase was significantly higher than that of male physicians (not or equivocal/slight/marked increase; 127 (71.3%)/39 (21.8%)/11 (6.2%) for male physicians vs. 49 (59.8%)/21 (25.6%)/12 (14.1%), respectively, for female physicians, p = 0.04).

## Factors for increasing domestic burden

Correcting for background differences, which are shown in Tables 1 and 2, based on multiple regression analysis, we investigated what factors were statistically related to an increased domestic burden during the first and second waves (Table 3). We found that gender (p<0.001), having dependent children/adult children (p<0.001), and the experience of treating patients with COVID-19 (p = 0.03) were significantly and positively related to an increased domestic burden.

**Table 3. Factors related to the domestic burden for physicians during the COVID-19 pandemic.**

| | Multivariate analysis | | | |
|---|---|---|---|---|
| | Regression coefficient | 95%CI | | P value |
| | | Lower | Upper | |
| Age | -0.005 | -0.103 | 0.091 | 0.904 |
| Gender (female) | 0.367 | 0.192 | 0.542 | <0.001[†] |
| Employment position | 0.012 | -0.050 | 0.074 | 0.698 |
| Dependent children/adult children | 0.329 | 0.183 | 0.476 | <0.001[†] |
| Pregnant | -0.082 | -0.548 | 0.384 | 0.728 |
| Elderly care | 0.010 | -0.328 | 0.350 | 0.949 |
| Experience of treating/caring for patients with COVID-19 (%) | 0.183 | 0.025 | 0.341 | 0.030[†] |

[†] P<0.05 was considered significant.

Adjusted R-squared: 0.1072, p-value: 0.000008248.

Table 4. Results of the questionnaire survey regarding childcare and family style.

| | | Total | Physician-fathers | Physician-mothers | p-value |
|---|---|---|---|---|---|
| | | (n = 154) | (n = 117) | (n = 37) | |
| Q. Who mainly performs childcare in your home? | Myself | 38 (24.7) | 3 (2.6) | 35 (94.6) | <0.01 |
| | My partner | 109 (70.8) | 108 (92.3) | 1 (2.7) | |
| | Grandparents | 2 (13.0) | 2 (1.7) | 0 (0.0) | |
| | Others/equivocal | 5 (3.2) | 4 (3.4) | 1 (2.7) | |
| Two- income (%) | Yes | 104 (67.5) | 68 (58.1) | 36 (97.3) | <0.01 |
| | No | 50 (32.5) | 49 (41.9) | 1 (2.7) | |

P<0.05 was considered significant.

Numbers in parentheses indicate percentages.

## Physician-mothers' dilemma: Work in and outside of home

We asked 154 physicians, (117 fathers and 37 mothers): "Who mainly performs childcare in your home?" The distribution of their answers showed significant differences between genders (p<0.01) (Table 4). Although 92.3% of physician-fathers answered, "my partner," 94.5% of physician-mothers answered, "myself." Meanwhile, a significantly larger number of physician-mothers were from two-income families, compared to physician-fathers in our institution (58.3% vs. 97.3%, p<0.01).

## The reasons for the increase in the domestic burden

Finally, we confirmed the reasons for the increase in the domestic burden, for male (n = 50) and female (n = 33) physicians who answered that it had increased (n = 83) (Table 5). According to single selectable questions, there was no significant difference between genders regarding the frequency of the answers. However, multiple selectable questions regarding childcare (response to the sudden closure of childcare service, learning support in the home, and mental support in the home) showed significant differences (p<0.001).

## Discussion

The impacts of the COVID-19 pandemic have been markedly changed the way in which humans perform their daily activities and go about their routine lives. Such impact has not

Table 5. Reasons for the increase in the domestic burden of the physicians.

| | | Total (n = 83) | Males | Females | p-value |
|---|---|---|---|---|---|
| | | | (n = 50) | (n = 33) | |
| Work associated with child care* | Total | 54 (65.1) | 31 (62.0) | 23 (69.7) | 0.47 |
| Work associated with child care** | Response to sudden closure of childcare service | 27 (32.5) | 10 (20.0) | 17 (51.5) | <0.001* |
| | Learning support in home | 23 (27.7) | 10 (20.0) | 13 (39.4) | |
| | Mental support in home | 26 (31.3) | 13 (26.0) | 13 (39.4) | |
| Work associated with elderly care | | 4 (4.8) | 2 (4.0) | 1 (3.0) | 0.81 |
| Hygienic housekeeping and/or cooking | | 26 (31.3) | 18 (36.0) | 7 (21.1) | 0.15 |
| Separation for infection control | | 4 (4.8) | 3 (6.0) | 1 (3.0) | 0.54 |
| Others | | 1 (1.2) | 0 (0.0) | 1 (3.0) | 0.21 |

*single selectable choice.

**multiple selectable choices.

P<0.05 was considered significant.

Numbers in parentheses indicate percentages.

been the same across the all citizens, the groups with the most vulnerable and marginalized having been affected differently due to the already existing social inequalities [3]. Especially, the pandemic has exacerbated the gender inequalities [4]. In this survey regarding work and lifestyle issues in the first and second waves of the COVID-19 pandemic, half of our hospital physicians responded. Even after correction for the background differences using multivariate analysis, being female, having dependent children, and treating COVID-19 patients were significantly related with an increased domestic burden. Interestingly, most physician-mothers answered that they had to take care of children by themselves. This was significantly different from the responses of the male physicians. Previous studies have reported on female physicians have struggle to balance their career and families [5–8]. Thus, the results showed that a significantly higher proportion of physician-mothers were caught in a dilemma between an increased home burden and clinical duties in the hospital than physician-fathers during the pandemic.

In Japan, most of the social support for a female doctor under the COVID-19 pandemic was limited. One reason why such limitation was coursed will be a character of this social crisis and unbalanced social structure in Japan. The lock-down for the control of the pandemic stopped traffics between the city and rural areas. Thus, if one parent treated COVID-19 patients in the rural hospital and stayed there, the other had to care for their child in their urban residence. In the Japanese social structure, especially in Hokkaido, most of the population (approximately 40%) is focused on the city. For these reasons, the need for babysitting was markedly elevated in the city area, and the Japanese government introduced economic support policies for babysitting services. However, a conventional babysitter has a risk of becoming an infection transmitter. The person who had both experiences of caring for children and prevention of infection is less, comparing to the need. Regarding parenting children under pandemic, this issue cannot easily find the answer and is a common topic globally. Under this context, we want to emphasize the contradiction that, although female doctors cannot take responsible positions in pre-pandemic society, they are asked to be responsible for this challenging issue. According to our survey, most female physicians, not males, had difficulty finding the ideal babysitter and caring for children under this pandemic.

During the COVID-19 pandemic, there has been an increasing in household responsibilities and care needs for children remaining at home with a marked decrease in options for stable or emergency childcare [9]. As well as in Japan, domestic responsibilities and household chores fall on women's shoulders much more heavily than on men's, even when the wife is a physician [10]. In the first and second waves in Japan, schools and nurseries were closed by the government, resulting in more children having to be cared for at home. This political decision resulted in a severe demand-supply gap on the childcare support system. As we showed, a total of 32.5% of physicians answered that they had to respond to the sudden closure of childcare services because support systems for childcare (babysitters, daycare, etc.) were not sufficient for the increased needs during the pandemic. This tendency was remarkable in physician-mothers, as 51.5% of them answered so. It is well known that there is a risk for increasing the burden on society when a political approach is adopted to control a social crisis (e.g., a pandemic). In a society with inequality, there is a tendency for the burdens to be placed on society as a whole, as occurred during the Great Depression and wars. This is inevitably more focused on those in weaker positions such as the socially vulnerable. In this pandemic, our hospital survey clearly showed that the burden was focused on physician-mothers. This fact suggests that we have to remedy this inequality to respond to the pandemic or another such social crisis.

Our study showed that female physicians in the hospital were younger, in lower employment positions, and had fewer dependent relatives. From this result, it could be inferred those female doctors cannot continue working with dependents, resulting in too short a working career to take a responsible position. These tendencies were not only typically shown in our

hospital but in the entire Japanese health care system. This is because Japanese society does not have a sufficient environment for female physician to continue their careers. Actually, according to the Global Gender Gap Report 2021 [11], Japan ranks 120th out of 156 countries in the comprehensive index of gender equality of health, education, economy, and politics, indicating that the social progress of females in this country lags far behind other nations.

## Limitations

This study has some limitations. First, there could be selection differences since not all of the physicians in our hospital, Hokkaido, or Japan, were included. For example, the proportion of persons who had great interest in this topic might be high. However, the tendencies of the backgrounds in Table 1 were approximately concordant with those of previous reports, as we discussed. Thus, there is a possibility that the effect of the differences is small in this study. However, it is considered that a larger well-designed study is needed to obtain stronger evidence for our results. Second, this study measured the subjective burden, not the quantitative one. To distinguish the quantitative from the qualitative, a comparative study with a labor fact-finding survey (e.g. working hours survey) and the subjective burden would be required. Third, balancing housework and hospital work may be a cause of an increased burden for females without children, but we were not able to examine whether this was due to an imbalance in the roles of the partners in the home, since we were not able to survey marital status. Fourth, the rate of the increase of home burdens among doctors with adult children was not statistically different from those with young children, and we have not been able to investigate the causes of this increase due to adult children. It was thought necessary to look more deeply into whether adult children live with parents, or they are financially independent.

## Conclusions

Our results show that a significantly higher proportion of physician-mothers were caught in a dilemma facing an increased home burden and clinical duties in the hospital than physician-fathers during the pandemic. As we showed, female doctors could have not continued their careers and take responsible positions in the same way as male doctors. This is a social risk in the timing of a crisis, such as a pandemic.

## Supporting information

**S1 Table. This is a result from questionnaire.**
(CSV)

**S1 File. This is an original questionnaire.**
(DOCX)

## Author Contributions

**Conceptualization:** Sachiyo Nishida.

**Data curation:** Masayuki Koyama, Hirofumi Ohnishi, Yoshihisa Tsuji.

**Formal analysis:** Masayuki Koyama, Hirofumi Ohnishi.

**Investigation:** Kanna Nagaishi, Masayo Motoya, Ayako Kumagai, Noriko Terada, Ai Kasuga, Narumi Kubota, Kotoe Iesato, Motonobu Kimizuka, Satsuki Miyajima.

**Methodology:** Yoshihisa Tsuji.

**Project administration:** Sachiyo Nishida, Kazufumi Tsuchihashi, Taiji Tsukamoto.

**Supervision:** Hirofumi Ohnishi, Eichi Narimatsu, Naoya Masumori, Kazufumi Tsuchihashi, Taiji Tsukamoto.

**Visualization:** Sachiyo Nishida.

**Writing – original draft:** Sachiyo Nishida.

**Writing – review & editing:** Yoshihisa Tsuji.

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
