## [Decision Letter · Decision Letter 0]

30 Apr 2021

PONE-D-21-10949

Dilemma of physician-mothers faced with an increased home burden and clinical duties in the hospital during the COVID-19 pandemic

PLOS ONE

Dear Dr. Nishida,

Thank you for submitting your manuscript to PLOS ONE. After careful consideration, we feel that it has merit but does not fully meet PLOS ONE’s publication criteria as it currently stands. Therefore, we invite you to submit a revised version of the manuscript that addresses the points raised during the review process.

We look forward to receiving your revised manuscript.

Kind regards,

Yoshihiko Hirohashi, M. D., Ph. D.

Academic Editor

PLOS ONE

Journal Requirements:

For more information on PLOS ONE's expectations for statistical reporting, please see https://journals.plos.org/plosone/s/submission-guidelines.#loc-statistical-reporting. Please update your Methods and Results sections accordingly

Additional Editor Comments :

Dear Authors,

In the study from authors Sachiyo Nishida et al. entitled ‘Dilemma of physician-mothers faced with an increased home burden and clinical duties in the hospital during the COVID-19 pandemic.’, the authors analyzed the survey of physicians regarding to treatments of patients and domestic burden under COVID-19 pandemic. The authors found that domestic burden have been increased especially in female and having child physicians and they suffer dilemma between increased home burden and clinical duties under COVID-19. This study raises significant problem of Japanese social system and this must be discussed intensively now. Some major and minor points have been raised by the reviewers. Please consider answering the comments.

Yours sincerely,

Reviewers' comments:

Reviewer's Responses to Questions

**Comments to the Author**

1. Is the manuscript technically sound, and do the data support the conclusions?

Reviewer #1: Yes

Reviewer #2: Yes

Reviewer #3: Yes

2. Has the statistical analysis been performed appropriately and rigorously? 

Reviewer #1: Yes

Reviewer #2: Yes

Reviewer #3: Yes

3. Have the authors made all data underlying the findings in their manuscript fully available?

Reviewer #1: Yes

Reviewer #2: Yes

Reviewer #3: Yes

4. Is the manuscript presented in an intelligible fashion and written in standard English?

Reviewer #1: Yes

Reviewer #2: Yes

Reviewer #3: No

5. Review Comments to the Author

Reviewer #1: I have reviewed the Manuscript ID PONE-D-21-10949 entitled "Dilemma of physician-mothers faced with an increased home burden and clinical duties in the hospital during the COVID-19 pandemic". The following concerns are listed below:

1) General comments

The authors evaluated the burdens on physicians in Sapporo Medical University, which managed by Hokkaido's prefectural government in Japan during coronavirus disease 2019 (COVID-19) pandemic, and concluded physician-mothers are caught in a dilemma between an increased home burden and clinical duties in the hospital.

It seems an unique and interesting and informative study during the COVID-19 pandemic.

2) Specific comments for revision

A) Major

1. Please mention the gender distribution of all 622 medical and dental physicians and researchers working in Sapporo Medical University.

2. The general background in your institution is well written in Introduction. However, please mention what kind of physician take care for COVID-19, how to share the work, and the role of researcher in your institution.

3. Please mention regarding the social support system in Japan in Discussion.

4. You have to provide proof number of ethical approval in the institution even though the board chairman’s judgement in your institution.

B) Minor

5. None

Reviewer #2: This paper investigates the effects of the coronavirus pandemic on physicians' lives at university hospitals having responsibility as a central role in medical care in Hokkaido, Japan. The COVID-19 impact resulted in an increase in the burden of housework of female doctors compared with male doctors, and a female doctor burdened the care of children in a working couple. Although the number of questionnaires is limited, the impact on physicians' daily lives under emergencies in groups with matching work environments has become clear, and it can be said that it is an important paper in Japan that is promoting work style reforms.

abstract

The purpose, method, and result are briefly described. Among the results, there is a risk factor for the increase in housework as treating patients, but it is inferr from the text that it is the treatment of Covid-19 patients. It is easier to use treating Covid-19 patient in the abstract. The conclusion suggests a warning about the gender gap and delayed systems of child care in Japan.

introduction

Information on Hokkaido, which is medical area of university hospitals and sudden school or childcare service closures during a pandemic, and the status of the accepting of COVID-19 patients at the university hospitals studied are shown.

Materials & Methods

The research design is excellent.

Outcome

Increase of domestic burden associated with the pandemic according to gender

The percentage of physicians who say they have increased house burden is 30% (including slightly and markedly), proving the big impact of COVID-19 emergencies on physicians' lives.

Factors for increasing domestic burden

Female doctors, having children, and COVID-19 patients care is a factor in the increase in domestic burden. These results suggest female physicians are more effected than male physician, because female physicians are always do house burden and child care.

Physician-mothers’ dilemma: work in and outside of home

In Table. 4, was there an investigation into the changes in those responsible for caring for children before and after the pandemic?

The reasons for the increase in the domestic burden

Of the 83 people who said they had to spent times for more housework, how many doctors had children? I think the impact of daycare centers and school closures is significant, but is there a link between children's age and the burden of housework?

Discussion

Line 252:treating patient to treating COVID-19 patient

Reviewer #3: I've pointed out some words that should be changed so check it out.

Conclusion: ”This suggests that we have to redress this inequality as soon as possible to avoid such problems in the event of another such social crisis, which could occur at any time. ”

This does not fit the content, so I think it should be changed as follows.

Other than this, please refer to the attached file.

In order for female doctors to continue their careers and take responsible positions in the same way as male doctors, it is necessary to improve the delay in gender equality in Japanese society.

6. PLOS authors have the option to publish the peer review history of their article (what does this mean?). If published, this will include your full peer review and any attached files.

Reviewer #1: No

Reviewer #2: No

Reviewer #3: No

---

## [Author Response · Author response to Decision Letter 0]

7 Jun 2021

PONE-D-21-10949

Dilemma of physician-mothers faced with an increased home burden and clinical duties in the hospital during the COVID-19 pandemic.

Dear Editor in Chief of PLOS ONE 

Thank you for giving us a chance to revise our manuscript. 

According to reviewer’s comments, we carefully revised the manuscript, which may be more informative for readers of the journal. Additionally, this study was approved by the institutional reviewing committee of Sapporo Medical University.

We hope that the editor and the reviewer will be satisfied with the current revised manuscript including our response to the comments. 

Thank you for your kind consideration and review of the manuscript.

In addition, I'm sorry, but I noticed an error and corrected the followings, which were not pointed out by the reviewers:

#1 L28 Division of General Medicine→ Department of General Medicine

#2 L204 *There was a 20s male doctor who did not answer this question only, so the total number of respondents was 263.

Sincerely Yours,

First author

Sachiyo Nishida, MD, PhD

Department of Urology, Sapporo Medical University, Sapporo, Japan

S-1 W-16, Sapporo-city, Hokkaido, Japan 060-8543

Tel; +81-11-611-2111

E-mail address: sachi@sapmed.ac.jp

Corresponding author

Yoshihisa Tsuji, MD, PhD,

Department of General Medicine, Sapporo Medical University, Sapporo, Japan

S-1 W-16, Sapporo-city, Hokkaido, Japan 060-8543

Tel; +81-11-611-2111

E-mail address: ytsuji@sapmed.ac.jp

---

## [Editor Report · Decision Letter 1]

10 Jun 2021

Dilemma of physician-mothers faced with an increased home burden and clinical duties in the hospital during the COVID-19 pandemic

PONE-D-21-10949R1

Dear Dr. Nishida,

We’re pleased to inform you that your manuscript has been judged scientifically suitable for publication and will be formally accepted for publication once it meets all outstanding technical requirements.

Kind regards,

Yoshihiko Hirohashi, M. D., Ph. D.

Academic Editor

PLOS ONE

Additional Editor Comments (optional):

The authors addressed concerns.
---

## [Editor Report · Acceptance letter]

17 Jun 2021

PONE-D-21-10949R1 

Dilemma of physician-mothers faced with an increased home burden and clinical duties in the hospital during the COVID-19 pandemic 

Dear Dr. Tsuji:

I'm pleased to inform you that your manuscript has been deemed suitable for publication in PLOS ONE. Congratulations! Your manuscript is now with our production department. 

Kind regards, 

on behalf of

Dr. Yoshihiko Hirohashi 

Academic Editor

PLOS ONE